# A new variant of the colistin resistance gene MCR-1 with co-resistance to β-lactam antibiotics reveals a potential novel antimicrobial peptide

Lujie Liang[1,2,3�], Lan-Lan Zhong[1,2,3�], Lin Wang[4�], Dianrong Zhou[1,2,3], Yaxin Li[1,2,3], Jiachen Li[1,2,3], Yong Chen[5], Wanfei Liang[1,2,3], Wenjing Wei[6], Chenchen Zhang[6], Hui Zhao[7], Lingxuan Lyu[1,2,3], Nicole Stoesser[8], Yohei Doi[9,10], Fang Bai[4]*, Siyuan Feng[1,2,3]*, Guo-Bao Tian[1,2,3]*

1 Advanced Medical Technology Center, The First Affiliated Hospital, Zhongshan School of Medicine, Sun Yat-sen University, Guangzhou, China, 2 Department of Immunology, School of Medicine, Sun Yat-Sen University, Shenzhen, China, 3 Key Laboratory of Tropical Diseases Control (Sun Yat-sen University), Ministry of Education, Guangzhou, China, 4 Shanghai Institute for Advanced Immunochemical Studies and School of Life Science and Technology, Shanghai Tech University, Shanghai, China, 5 School of Laboratory Medicine, Chengdu Medical College, Chengdu, China, 6 Center for Tuberculosis Control of Guangdong Province, Guangzhou, Guangdong, China, 7 Laboratory Medicine, Guangdong Provincial People's Hospital, Guangdong Academy of Medical Sciences, Guangzhou, Guangdong, China, 8 Modernising Medical Microbiology, Nuffield Department of Medicine, University of Oxford, Oxford, United Kingdom, 9 University of Pittsburgh School of Medicine, Pittsburgh, Pennsylvania, United States of America, 10 Department of Microbiology, Fujita Health University School of Medicine, Aichi, Japan

८ These authors contributed equally to this work.
* baifang@shanghaitech.edu.cn (FB); fsy0593@163.com (SF); tiangb@mail.sysu.edu.cn (G-BT)

**Data Availability Statement:** All relevant data are within the paper and its Supporting Information files. The custom code used in this research for proteomics analysis, MCR proteins homology

## Abstract

The emerging and global spread of a novel plasmid-mediated colistin resistance gene, *mcr-1*, threatens human health. Expression of the MCR-1 protein affects bacterial fitness and this cost correlates with lipid A perturbation. However, the exact molecular mechanism remains unclear. Here, we identified the MCR-1 M6 variant carrying two-point mutations that conferred co-resistance to β-lactam antibiotics. Compared to wild-type (WT) MCR-1, this variant caused severe disturbance in lipid A, resulting in up-regulation of L, D-transpeptidases (LDTs) pathway, which explains co-resistance to β-lactams. Moreover, we show that a lipid A loading pocket is localized at the linker domain of MCR-1 where these 2 mutations are located. This pocket governs colistin resistance and bacterial membrane permeability, and the mutated pocket in M6 enhances the binding affinity towards lipid A. Based on this new information, we also designed synthetic peptides derived from M6 that exhibit broad-spectrum antimicrobial activity, exposing a potential vulnerability that could be exploited for future antimicrobial drug design.

analysis and homology protein modelling were deposited in https://github.com/LiangLujie/MCR-1_new_variant and https://github.com/Wang-Lin-boop/AutoMD. The protein structure of MCR-1 and its lipid A binding is presented in S1 Supplementary PDB file.

**Funding:** This work was supported by the National Natural Science Foundation of China (grant number 81830103, 82061128001 and 82325033 to G-BT, grant number 82002173, 82272378 to SF), Natural Science Foundation of Guangdong Province (grant number 2017A030306012 to G-BT, grant number 2023A1515012392 to SF), Scientific and Technological Planning Project of Guangzhou City (grant number SL2022A04J01941 to SF). The funders had no role in study design, data collection and analysis, decision to publish, or preparation of the manuscript.

**Competing interests:** The authors declare no conflicts of interest.

**Abbreviations:** AMP, antimicrobial peptide; CFU, colony-forming unit; CRE, carbapenem-resistant Enterobacteriaceae; GO, Gene Ontology; IM, inner membrane; LB, Luria–Bertani; LPS, lipopolysaccharide; MCRPE, mcr-1-positive Enterobacterales; NPN, n-phenyl-1-napthylamine; OM, outer membrane; PE, phosphatidylethanolamine; PEA, phosphoethanolamine; PI, propidium iodide; PG, peptidoglycan; PPI, protein–protein interaction; SEM, scanning electron microscopy; sfGFP, super-folded green fluorescent protein; TEM, transmission electron microscopy.

# Introduction

Colistin/Polymyxin is a last-resort antibiotic against infections caused by highly drug-resistant bacteria, particularly carbapenem-resistant *Enterobacterales* [1]. However, the emergence of a novel plasmid-mediated colistin resistance gene named *mcr-1* in 2016 threatened the clinical effectiveness of colistin. Since then, instances of *mcr-1*-positive *Enterobacterales* (MCRPE) have been detected from various sources (including livestock, humans, animal food products, and the environment) and have disseminated globally, spreading to >40 countries in 5 continents [2–4]. Colistin has been used as an additive in livestock feed to promote growth and prevent infection in China since the 1980s, and the correlation between the spread of *mcr-1* and colistin use in animal husbandry has been observed [5]. To prevent the continued spread of plasmid-borne *mcr-1*, the Chinese government banned the use of colistin as animal feed additive in 2017. Such policy resulted in remarkable reductions in the production as well as sale of colistin sulfate premix [6] and a subsequent drastic decline in *mcr-1* prevalence [7]. Nonetheless, a low prevalence of *mcr-1* was still detected among inpatients, likely associated with the approval of colistin for clinical use in China [5,6,8,9]. Furthermore, increased co-existence of *mcr-1* and carbapenemase genes after the clinical introduction of polymyxin was recently reported [10]. Hence, there is an urgent need to develop new strategies to eliminate the spread of *mcr-1* and prolong the use of colistin/polymixin as the last-resort antibiotic against carbapenem-resistant bacteria that carry *mcr-1*.

By neutralizing the overall negative charge of the bacterial surface, MCR-1 renders the bacteria resistant to colistin, which functions as an inner-membrane metalloenzyme with its active site exposed to the periplasmic space [11,12]. MCR-1 comprises 5 predicted membrane-spanning α-helixes, forming an insoluble domain that anchors in the cytoplasmic membrane. This domain is connected to a soluble catalytic domain, which extends into the periplasm through a flexible linker region [13,14]. The catalytic domain of MCR-1 is highly homologous with that of the *Neisseria meningitidis* phosphoethanolamine (PEA) transferase EptA [11,15]. Both of them possess a conserved zinc-binding pocket for conferring colistin resistance. The MCR-1–mediated lipid A modification involves 2 steps: (i) phosphatidylethanolamine (PE) hydrolysis into diacylglycerol resulting in MCR-1-bound PEA; and (ii) transfer of the PEA moiety onto lipid A [16]. PE is the major lipid component of inner membrane (IM), flipped by MsbA [17,18]. While the PE-interacting cavity has been assigned [19], the precise mechanism for the interplay between MCR-1 and lipid A remains unknown.

The lipid A moiety of lipopolysaccharide (LPS), the primary component of outer membrane (OM) in gram-negative bacteria, is critical to bacterial physiology and survival [20,21]. Lipid A is synthesized on the cytoplasmic surface of the IM through a well-conserved pathway involving 9 enzymes. After attaching the core oligosaccharide, lipid A is flipped to the outer leaflet of the IM by the ABC transporter MsbA. Subsequently, O-antigen polymers may added in some but not all gram-negative bacteria [22]. LPS molecules are transported from the IM to the OM and distributed asymmetrically on the outer leaflet of the OM [23], a process requiring maintenance of lipid asymmetry (Mla) system. The proper distribution of lipid A is crucial for maintaining the mechanical integrity of the bacterial OM [24,25]. The interaction between MCR-1 and lipid A appears to compromise bacterial physiology. Various studies have demonstrated that MCR-1 leads to reduced bacterial fitness [26–29], even for isolates collected from hospitals [30]. The reduced *mcr-1* prevalence after the ban on colistin-added fodders is believed to be associated with the MCR-1-mediated fitness cost in the absence of antibiotic selective pressure [6,7,9]. Our recent research has suggested that MCR-1 induces membrane lipid remodeling, leading to a compromise in the OM integrity as well as bacterial viability [31]. These observations raise the possibility that MCR-1 possesses a potential lipid-binding

pocket that disrupts the distribution of lipid A in the OM. However, the precise molecular mechanism by which MCR-1 disrupts lipid homeostasis remains to be fully elucidated.

Using our previous high-throughput library of *mcr-1* mutants, we characterize here an MCR-1 variant named M6, which possesses 2 mutations (P188A and P195S) within the linker domain. M6 expression resulted in co-resistance towards β-lactam antibiotics. Meanwhile, the expression of M6 caused membrane lipid A perturbation, resulting in a growth defect and up-regulation of the LDTs pathway. Moreover, we identified a lipid A loading pocket located within the linker domain of MCR-1, where the 2 point mutations in M6, potentially increasing the lipid A binding affinity. Synthetic peptides derived from the M6 linker domain revealed antimicrobial activity, unraveling a potentially new strategy for eliminating drug-resistant bacteria.

## Results

### MCR-1 variant M6 induces co-resistance towards β-lactam antibiotics

Given that MCR-1 disrupts OM integrity [31,32] facilitating entry of various antibiotics, we hypothesized that specific mutations in *mcr-1* could abolish this phenotype. We tested this hypothesis using our previously established *mcr-1* mutant library, which encompassed 171,769 mutation genotypes [33], covering 99.96% (4858/4860) possible single-nucleotide mutations of *mcr-1*. Screening was performed at antibiotic concentrations below and above the minimal inhibitory concentration (MIC, 0.8–2× MIC) to distinguish mutants with low- and high-level resistance (S1A Fig and S1 Table). By counting the colony-forming unit (CFU) values evalu-ated from plates containing aminoglycosides (streptomycin, SM), tetracyclines (tetracycline, TET), quinolones (nalidixic acid, NAL), or glycopeptides (vancomycin, VAN), we observed that some colonies from MCR-1 library could grow on plates containing penicillin (ampicillin, AMP), cephalosporin (ceftazidime, CAZ), or carbapenem (imipenem, IMP) (S1B Fig). More-over, we confirmed that the MCR-1 library exhibited higher viability than the control strains after exposure to β-lactams (S1C Fig and S2 Table). Next, 50 isolates of the MCR-1 library growing on the plates containing 2 × MIC CAZ or AMP were selected, and the *mcr-1* geno-types of the selected isolates were identified by Sanger sequencing (S3 Table). Subsequently, we conducted MIC assays to evaluate the susceptibility of the selected strains upon CAZ, AMP, and FOX. Several mutants displayed low-level resistance to AMP or CAZ (S3 Table). We identified a variant named M6, carrying 2 mutations (P188A and P195S) localized at an α-helix of the linker domain (Fig 1A and S3 Table), a region of low conservation among the pro-tein structures of the MCR family (S2 Fig). Compared to MCR-1-expressing cells, this mutant displayed increased resistance towards several β-lactams while maintaining resistance towards colistin (Fig 1B and Table 1), but not to other types of antibiotics (S3A Fig). This phenotype was further confirmed by regulating M6 expression with MCR-1 native promoter (S4A Fig). In addition, MCR-1 with either P188A or P195S abolished the co-resistance phenotype (S3B Fig), indicating that simultaneous mutation in both residues was required.

The expression of M6 had a significant impact on bacterial growth compared to MCR-1 (Fig 1C). We speculated that the phenotypic co-resistance may be related with an increase in membrane permeability conferred by M6. To confirm this, we assessed whether M6 confers resistance to membrane disrupting agents like SDS/EDTA. As observed before [31], MCR-1-expressing cells displayed increased sensitivity to detergents compared with control, whereas M6 expression partially recovered host permeability (Fig 1D). We further quantified the level of membrane permeability by n-phenyl-1-napthylamine (NPN) and propidium iodide (PI) uptake assays. Comparing with control, increased NPN uptake was observed in both the MCR-1 or M6 expressing cells (Fig 1E). Moreover, M6 expression significantly reduced the

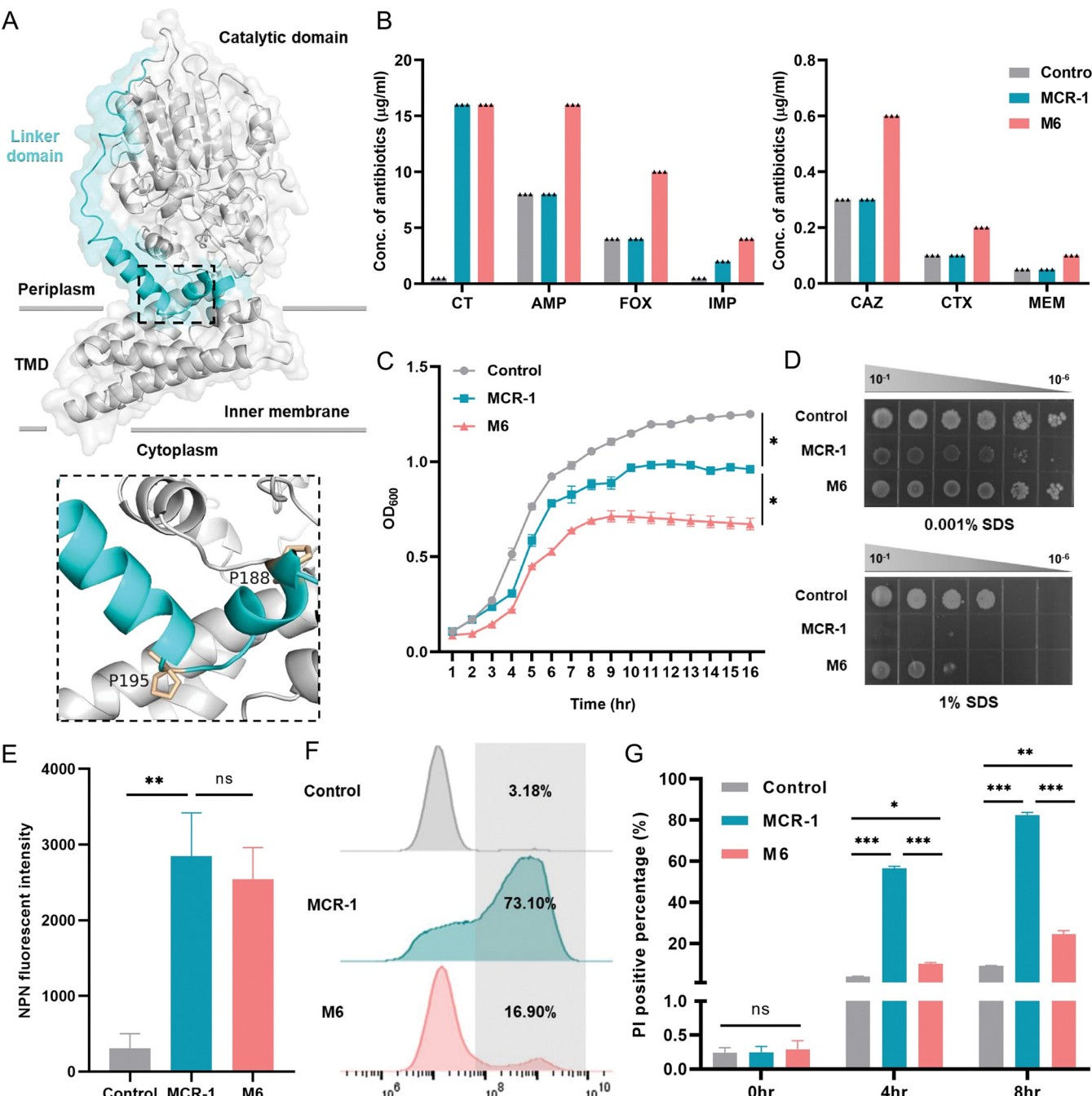

**Fig 1. An MCR-1 mutant M6 (harboring P188A+P195S) exhibited reduced sensitivity towards β-lactam antibiotics. (A)** MCR-1 protein structure in cartoon. The linker domain (blue) and approximate membrane boundaries are indicated. The mutated residues P188 and P195 are highlighted in orange. Close-up view of 2 mutated residues is represented. TMD, transmembrane domain. **(B)** The logarithmic phase cultures of *E. coli* BW25113 carrying empty vector, M6, or MCR-1 were collected to evaluate the sensitivity towards CT and β-lactam antibiotics (AMP, FOX, IMP, CAZ, CTX, and MEM) by agar dilution MICs tests. Each triangle represents an independent experiment. And the experiments were performed 3 times with same results. **(C)** Growth curves of *E. coli* BW25113 carrying empty vector, M6, or MCR-1. The y-axis shows optical densities at 600 nm ($OD_{600}$) of broth cultures, x-axis shows period of growth (h). **(D)** Efficiency of plating assays on LB agar plates containing 1% SDS and 1 mM EDTA or 0.001% SDS and 1 mM EDTA. Ten-fold serial dilution of indicated cultures was inoculated onto the agar plates. **(E)** The OM integrity of strains carrying M6 or MCR-1 was determined by measuring NPN uptake during logarithmic phase. And the fluorescent signal for each sample was immediately monitored with a microplate reader at an excitation wavelength of 350 nm and emission wavelength of 420 nm after staining. **(F, G)** The IM permeability of cells carrying MCR-1 or M6 was evaluated by PI staining assay. Overnight cultures were subcultured into fresh LB broth at a ratio of 1:100 and induced with 0.2% arabinose to express MCR-1 or M6. After induction for 4 h and 8 h, stationary and late-stationary phase cultures were collected, respectively, followed by staining with PI dye for 15 min. The PI-positive proportion was determined by flow cytometry and analyzed by FlowJo version 10 software. All the above-described experiments were performed thrice with similar results. Error bars indicate

standard errors of the means (SEMs) for 3 biological replicates. A two-tailed unpaired $t$ test was performed to determine the statistical significance of the data. ns, no significant difference; *, $P < 0.1$; **, $P < 0.01$; ***, $P < 0.001$. The raw data underlying this figure can be found in S1 Data. IM, inner membrane; LB, Luria–Bertani; NPN, n-phenyl-1-napthylamine; OM, outer membrane; PI, propidium iodide.

number of PI-positive cells during stationary phase compared to MCR-1-expressing cells (Fig 1F and 1G). Similar phenotypes were observed when M6 expression was controlled by the MCR-1 native promoter (S4B–S4E Fig). These observations suggested that the co-resistance phenotype mediated by M6 could be related to membrane damage. Overall, these results revealed that M6 conferred a co-resistance towards β-lactam antibiotics via an unknown mechanism enhancing membrane integrity.

## M6 expression causes a lipid A defect in the outer membrane

We next investigated the molecular mechanism by which M6 expression caused co-resistance towards β-lactams. OM assembly defects can activate the LDT pathway, which mediates peptidoglycan (PG) layer remodeling and resistance against β-lactams [34–36]. Using scanning electron microscopy (SEM), we found that MCR-1-expressing cells displayed wrinkles. In contrast, M6-expressing cells exhibited significant morphological changes, including folded membrane stacks and holes at the bacterial poles (Fig 2A). Transmission electron microscopy (TEM) revealed that M6 expression induced noticeable interspace within the periplasm at one pole of the cell (Fig 2B), indicating cytoplasmic shrinkage. To further quantify the level of bacterial membrane perturbation, we constructed a two-fluorescence reporter system to distinguish bacterial periplasm and cytoplasm, as described [37]. In this system, super-folded green fluorescent protein (sfGFP) was fused with the signal peptide of DsbA and localized in the periplasm, while mCherry was constitutively expressed in the cytoplasm (S5A Fig). Stationary phase cultures were collected for analysis. M6-expressing bacteria exhibited a higher proportion of cells accumulating sfGFP at bacterial pole(s) (90.7%, S5B Fig) than MCR-1 containing bacteria (61.3%). The formation of sfGFP foci at bacterial pole, consistent with IM shrinkage [37], suggested that the expression of M6 contributed to more severe membrane perturbation compared to MCR-1. We also investigated the impact of M6 upon membrane potential by using a sensitive bacterial membrane voltage $V_m$ sensor, which allow single-cell recording of bacterial $V_m$ dynamics in live cells. The results show that M6 confers increased membrane voltage (S6A and S6B Fig). Since membrane permeability is a critical determinant for membrane potential, this result also confirms a disruption of membrane homeostasis caused by M6 expression.

We employed label-free quantitative proteomics to profile proteome changes in M6-bearing $E.\ coli$. In total, 2,287 proteins were identified, covering 55.5% of the $E.\ coli$ BW25113 proteome (S7A Fig and S4 Table). With our significant criteria (|log2-fold change (M6/MCR-1)|≥ 1 and $p\ value$ <0.05), 66 differentially expressed proteins were identified (41 up-regulated and 25 down-regulated). Gene Ontology (GO) analysis revealed a significant enrichment in biological events related to bacterial cell membrane and envelope space (S7B Fig and S5 Table). In

**Table 1. β-Lactam antibiotics MICs of M6-bearing $E.\ coli$.**

| Strains | MICs (µg/ml) | | | | | |
|---|---|---|---|---|---|---|
| | CAZ | AMP | PIP | CTX | FOX | CT |
| M6-bearing $E.\ coli$ BW25113 | ≥0.6 | 16 | 4 | 0.2 | ≥10 | ≥16 |
| MCR-1-bearing $E.\ coli$ BW25113 | 0.2 | 8 | 4 | 0.05 | 4 | ≥16 |
| Control carrying empty vector | 0.2 | 8 | 4 | 0.05 | 4 | 0.5 |

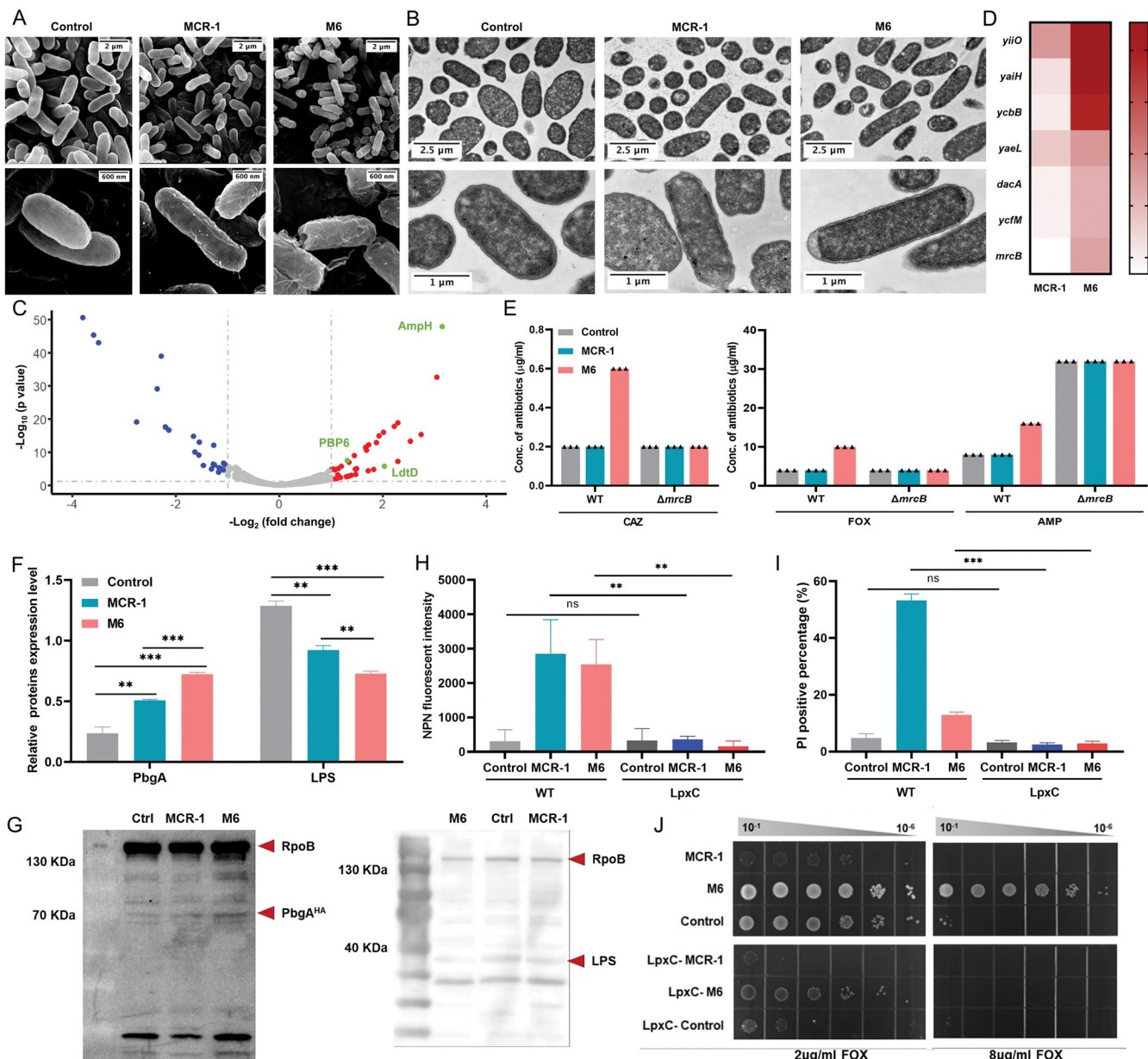

**Fig 2. M6 activated LDTs pathway via the induction of severe lipid disorders. (A, B)** SEM and TEM micrographs of *E. coli* BW25113 carrying MCR-1 or M6. Overnight cultures of the above strains were subcultured into fresh LB broth at a ratio of 1:100 and induced with 0.2% arabinose for 2 h before sample preparation. **(C)** The volcano diagram of the differentially expressed protein between MCR-1-bearing cells and M6-bearing cells determined by label-free quantitative proteomics. The differential expression threshold was set as log$_2$ fold change >1 and *p* value <0.05. Red dots represent significantly up-regulated genes, while blue dots represent significantly down-regulated genes. The LDTs pathway-related proteins are highlighted in green. **(D)** The transcriptional level of differentially expressed genes was measured by q-PCR, which was normalized to the transcript level of the housekeeping gene *rpoB* and quantified with ΔΔCT analysis. The heatmap represents the fold change in target gene transcriptional levels for MCR-1-expressing cells or M6-expressing cells compared to that of the empty plasmid control. **(E)** Evaluation of the impact of *mrcB* on the M6-mediated co-resistance phenotype. The antibiotic susceptibility of WT or Δ*mrcB* carrying empty plasmid (control), MCR-1 or M-6 was determined by agar dilution MIC tests. Each triangle represents an independent experiment. And the experiments were performed 3 times with same results. **(F, G)** The logarithmic phase cultures of *E. coli* BW25113 carrying empty vector, M6, or MCR-1 were collected. And the cellular levels of PbgA and LPS of indicated strains were determined by western blot after cell lysis with ultrasonication. **(H)** Evaluation of the impact of LpxC overexpression on M6-mediated OM permeability. The OM integrity was determined by measuring NPN uptake of *E. coli* during logarithmic phase. **(I)** Evaluation of the impact of LpxC overexpression on M6-mediated IM permeability. The IM integrity was evaluated by PI staining assay. **(J)** Assessment of the impact of LpxC on the M6-mediated co-resistance phenotype. The FOX MICs of WT or LpxC overexpression strains carrying empty plasmid (control), MCR-1 or M6 were determined by agar dilution MICs tests. All the above-described experiments were performed 3 times with similar results. Error bars indicate standard errors of the means (SEMs) for 3 biological replicates. A two-tailed unpaired *t* test was performed to determine the statistical significance of the data. ns, no significant difference; **, *P* < 0.01; ***, *P* < 0.001. The raw data underlying this figure can be found in S1 Data. IM,

inner membrane; LB, Luria–Bertani; NPN, n-phenyl-1-napthylamine; OM, outer membrane; PI, propidium iodide; SEM, scanning electron miscopy; TEM, transmission electron microscopy.

KEGG analysis, an augmented process of PG layer synthesis was observed among the top enriched processes (S7C Fig and S6 Table). Proteins of LDTs pathway, such as AmpH, PBP6, and LdtD, were significantly up-regulated (Fig 2C). A protein–protein interaction (PPI) analysis further revealed, in addition to AmpH and PBP6, LpoB and PBP1B might also play crucial roles in the regulation of LDTs pathway (S7D Fig). Consistent with these findings, the transcriptional levels of genes responsible for the envelope stress response and LDTs pathway, including *yiiO* (gene encoding CpxP), *yaiH* (gene encoding AmpH), *ycbB* (gene encoding LdtD), and *yaeL* (gene encoding RseP), were significantly up-regulated (Fig 2D). This result indicated that M6 primarily influenced the LDTs pathway.

LdtD and its partner proteins, PBP1B and LpoB, are essential for the activity of LDTs pathway as well as PG layer remodeling [36]. Since LdtD-mediated PG layer remodeling enables the survival of gram-negative bacteria challenged by β-lactams [38], we explored whether M6 affects the LDTs pathway. In the presence of 3.75 mM $CuSO_4$, an inhibitor of LDTs [39], the M6-expressing strain exhibited increased susceptibility towards CAZ and AMP compared to the untreated group (S8A Fig). Furthermore, the deficiency of *mrcB* (gene encoding PBP1B) reduced the CAZ, AMP, and FOX MICs of M6-expressing strain (Fig 2E). However, deficiency in *ycbB* (gene encoding LdtD) or *ycfM* (gene encoding LpoB) had no effect on susceptibility of M6 towards β-lactams (S8B Fig), indicating the dominant role of PBP1B in the M6-mediated phenotypic co-resistance. Additionally, only the absence of PBP1B (i.e., Δ*mrcb*) in M6-expressing cells decreased the resistance to SDS detergent (S8C Fig). No significant difference in NPN uptake was observed between M6-expressing WT *E. coli* and *mrcB* null mutant (S8D Fig). However, the percentage of PI-positive cells in M6-bearing Δ*mrcB* mutant was significantly higher than that in the WT strain (S8E and S8F Fig). These results suggested that the LDTs pathway contributes to the barrier function of IM. To further verify the IM integrity of the M6-bearing strain, spheroplasts generated by removing the OM and PG layer were confirmed by microscopy (S9A Fig). Remarkably, the M6-bearing spheroplasts exhibited increased PI-positive proportion than the whole cells (S9B and S9C Fig), demonstrating reduced IM integrity of M6-expressing cells. Overall, these results indicated that PBP1B was essential for phenotypic co-resistance and IM barrier function in M6-bearing *E. coli*.

Because defective LPS assembly activates LDTs pathway in *E. coli* [36], we hypothesized that the up-regulation in LDTs induced by M6 could be due to lipid A disruption. Therefore, we determined the expression level of PbgA, a periplasmic protein that senses lipid A levels and regulates the content of LPS in the OM [40]. The PbgA level in M6-expressing cells was higher than that in MCR-1-expressing cells (Fig 2F and 2G). Moreover, the overall and IM levels of LPS in M6-expressing cells was significantly lower compared to MCR-1-expressing cells (Figs 2F, 2G and S10A). These results demonstrated that the expression of M6 led to significant reduction in cellular lipid A level, which might be responsible for the increased permeability of both OM and IM. We next generated strains overexpressing LpxC (an essential enzyme for lipid A biosynthesis), which led to an increase in the LPS expression (S10B Fig). Consistently, permeabilization caused by MCR-1 or M6 were reversed with the compensation of LpxC (Fig 2H and 2I). In addition, the overexpression of LpxC abolished the co-resistance phenotype conferred by M6 (Fig 2J), highlighting the connection between membrane lipid A level and co-resistance to β-lactam antibiotics in M6-bearing strain. Taken together, these results demonstrate that a lipid A defect induced by M6 expression conferred reduced sensitivity to β-lactam antibiotics.

## Uncovering an essential lipid A binding pocket

We hypothesized that the linker region, where the M6 point mutations occurred, might play a significant role in the enzyme function of MCR-1. To test this idea, we generated a MCR-1 mutant lacking P188-P195 segment in the linker domain. As expected, ΔP188-P195 mutation abolished the colistin resistance of *mcr-1*-bearing strain (S11A Fig), resuming the susceptibility to β-lactams (S11B Fig) as well as SDS tolerance (S11C Fig). To investigate whether this region might serve as the potential lipid A loading pocket of MCR-1, we conducted molecular docking simulations with native lipid A conformations and MCR-1. A putative lipid A binding pocket with a total of 18 residues that localized at the linker domain was proposed (Fig 3A). Notably, 4 regions seemed to be critical for anchoring lipid A within the loading pocket (Figs 3B and S12). To further validate the significance of this predicted pocket upon MCR-1, we conducted structure-guided and site-directed mutagenesis. MICs assays revealed that these mutants played different roles in the context of colistin resistance (Fig 3B). For example, a hydrogen network was observed within K211, K204, and a phosphorate group of lipid A, and the two-point mutation K211A+K204A in MCR-1 significantly increased the susceptibility towards colistin. Additionally, a large hydrophobic region consisting of L64, I65, L68, L69, I165, and I168 likely played a role in stabilizing the flexible fatty acid tails of lipid A. The L64A +I65A+L68A+L69A mutations inactivated the function of MCR-1. We next investigated whether mutation in the predicted lipid A binding pocket might mitigate the membrane permeabilization. The mutations that abolished the colistin resistance activity of MCR-1 also restored both the OM and IM integrity of the bacterial host to the same as those of control (Fig 3B). Moreover, the colistin susceptibility phenotype for these mutants was highly correlated with the integrity of the IM ($\rho = 0.93$, $P = 0.00026$) or OM ($\rho = 0.93$, $P = 0.00026$) (Fig 3C). In short, these results strongly suggested that the linker region of MCR-1, where the P188A +P195S mutations occurred, serves as the lipid A binding pocket, governing phenotypic colistin resistance and bacterial membrane integrity.

To further investigate the impact of P188A+P195S mutations upon lipid A binding pocket, we conducted molecular dynamics simulations over M6 and MCR-1 to obtain probable lipid A binding conformations and evaluate their binding stabilities (S13A Fig). The binding affinities between lipid A and WT MCR-1 or M6 were evaluated by using the MM-GBSA method, and the results showed that M6 exhibited higher affinity towards lipid A than the WT MCR-1 (Fig 3D). Moreover, as illustrated in S13B and S13C Fig, the frequencies of the interactions between lipid A and specific residues of M6 were different from those observed with MCR-1. For example, the P188A+P195S mutations increased the interaction frequency between K204 and lipid A which was not observed in WT MCR-1 (S13D Fig). To further confirm that this pocket facilitates the lipid A binding affinity, we employed synthetic peptides encompassing the pocket of MCR-1 and M6, named peptide MCR-1 (KPLRSYVNPIMPIYSV) and peptide M6 (KALRSYVNSIMPIYSV), to interact with LPS. The dissociation constant ($K_D$) values were approximately 12.46 nM and 4.75 nM (Fig 3E and 3F), respectively. This result further demonstrated the presence of a potential lipid A binding pocket at the linker domain of MCR-1 and confirmed the enhanced binding affinity towards lipid A of the mutated pocket. Finally, to investigate the impact of P188A+P195S mutations upon lipid A distribution, we undertook MALDI-TOF to analyze the abundance of modified or unmodified lipid A purified from MCR-1 or M6-bearing cells. We found that higher abundance of unmodified lipid A was detected than that of the modified lipid A in the spheroplasts expressing MCR-1, which was in line with distribution of lipid A in the whole cells of MCR-1-bearing strains (S13E Fig). Inversely, the abundance of modified lipid A was higher than that of the unmodified lipid A in the M6-bearing spheroplasts. Together, these results indicated that the P188A+P195S

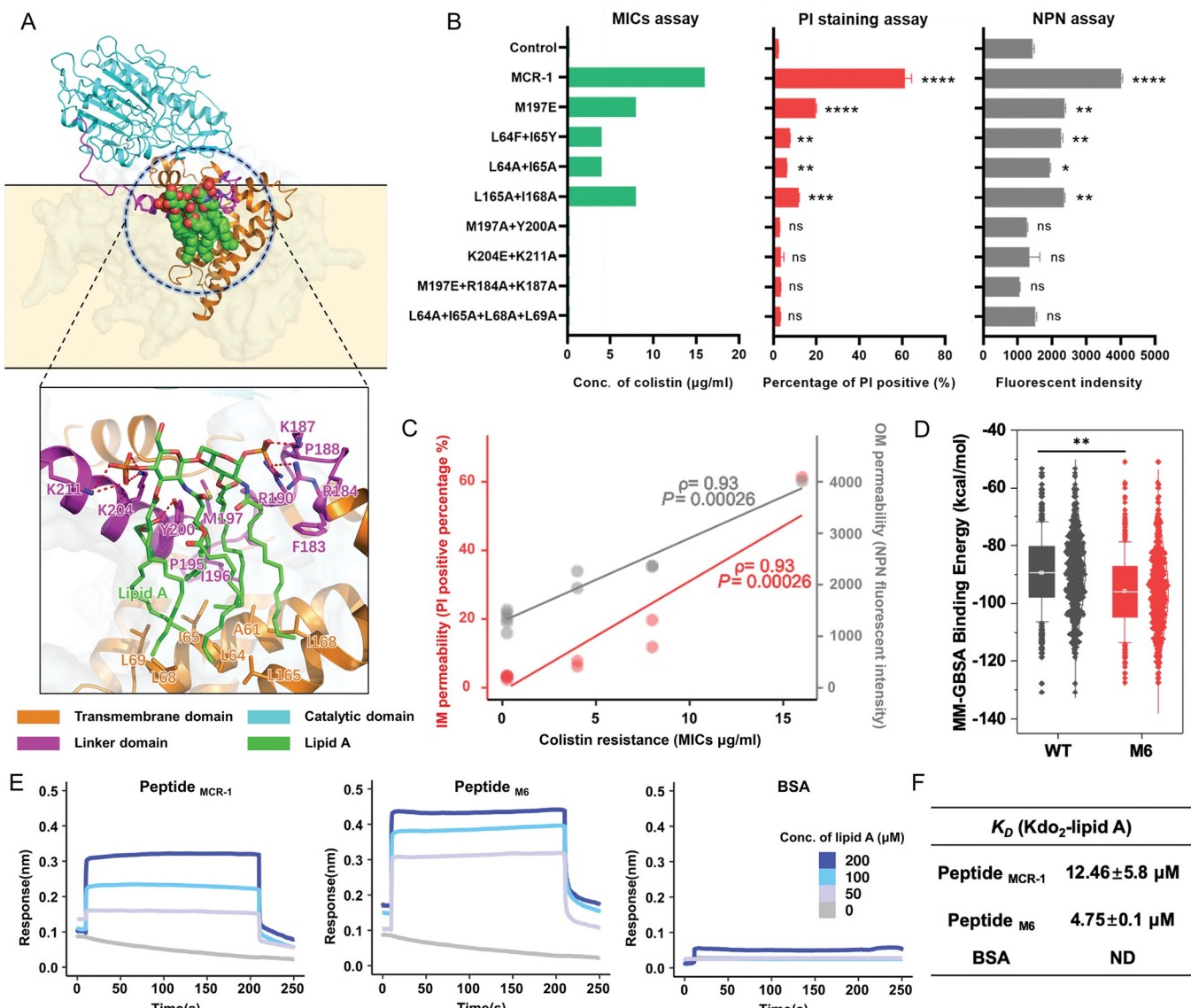

**Fig 3. Discovery and functional definition of the lipid A loading pocket in MCR-1.** (**A**) Protein structure of MCR-1 and close-up view of the LPS-binding pocket at the linker domain. The catalytic domain, linker domain, and transmembrane domain are in cyan, magenta, and orange, respectively, with LPS is shown as green sticks and salt bridges for interaction as red dashes. (**B**) Verification of the key residues in the lipid A binding cavity that interacts with lipid A. To evaluate the influence of mutations upon colistin resistance activity of MCR-1, the colistin MICs of the *E. coli* trains carrying indicated MCR-1 mutants were determined by agar dilution MICs tests. In addition, logarithmic phase cultures of target strains were applied to evaluate the permeability of the OM and IM by NPN uptake assay and PI staining assay, respectively. (**C**) Evaluation of the correlation between membrane integrity and colistin resistance. ρ is Spearman's rank correlation coefficient. The related *P* value and regression line (blue) are shown. (**D**) Estimation of the free energy of binding between lipid A and WT MCR-1 or M6. The results show that the M6 mutant has a lower binding free energy compared to WT, where the binding free energy was calculated using the MM-GBSA for the last 200 ns trajectory after the protein-ligand complex reached equilibrium. (**E, F**) The affinity between lipid A substrate and linker domain of MCR-1/M6 was determined by biolayer interferometry. The synthetic peptides derived from the linker domain of WT MCR-1 or M6 (named peptide MCR-1 and peptide M6, respectively) were coated on the surface of streptavidin (SA) biosensors. Target peptides were bound to Kdo$_2$-lipid A in series of concentrations (150, 100, 50, 25, and 10 μM) with 300 s association and dissociation steps. Assays were performed in triplicate on an Octet Red384 (Sartorius). All the above-described experiments were performed 3 times with similar results. Error bars indicate standard errors of the means (SEMs) for 3 biological replicates. A two-tailed unpaired *t* test was performed to determine the statistical significance of the data. ns, no significant difference; *, $P < 0.1$; **, $P < 0.01$; ***, $P < 0.001$; ****, $P < 0.0001$. The raw data underlying this figure can be found in S1 Data. IM, inner membrane; LPS, lipopolysaccharide; NPN, n-phenyl-1-napthylamine; OM, outer membrane; PI, propidium iodide.

mutations enhanced the binding affinity towards lipid A of MCR-1 and potentially resulted in accumulation of modified lipid A on the bacterial IM.

## Synthetic peptides derived from the M6 variant achieve broad-spectrum activity

Inspired by previous research from Thomas and colleagues [40], we postulated that peptides derived from the lipid A binding cavity of MCR-1 or M6 could have antimicrobial activity. To investigate this, we designed two 24AA short peptides derived from the lipid A binding pocket of WT MCR-1 and M6, termed 24AA-WT (KPLRSYVNPIMPIYSVGKLASIEY) and 24AA-2M (KALRSYVNSIMPIYSVGKLASIEY), respectively. While peptide 24AA-WT showed no antimicrobial activity, peptide 24AA-2M exhibited activity against *E. coli* ATCC 25922 and maintained its activity during the first 6 h (Fig 4A). However, after 24 h, no difference in bacterial viability occurred between 24AA-2M and 24AA-WT, indicating inactivation of the synthetic peptide. To enhance the stability of the indicated peptide, we generated a derivative of 24AA-2M by adding a five-tryptophan (5W) tag at the C-terminus, named 19AA-2M-tag (KALRSYVNSIMPIYSVGKLWWWWW), since such modification aimed to increase peptide stability and protect against protease activity [41–44]. Promising results were obtained with the 19AA-2M-tag, which showed potent activity against *E. coli* ATCC 25922 and maintained its activity for at least 24 h. SEM revealed that both 19AA-2M-tag and 24AA-2M induced severe OM shrinkage in *E. coli* (Fig 4B). Moreover, by labeling the both indicated peptides with FITC fluorescent dye at the C-terminal, we found that the FITC signal was overlapped with the signal of membrane specific dye FM4-64 (S14 Fig), which was in line with the previous observations of the antimicrobial peptides functioned by interacting with bacterial membrane [45]. Importantly, 19AA-2M-tag treatment exhibited higher antimicrobial activity compared to 24AA-2M-tag (Fig 4C and 4D), and no significant membrane lytic activity was observed upon mouse red blood cells treated with 19AA-2M-tag, indicating that it might be a promising antimicrobial peptide (S15 Fig).

We tested the susceptibility of 19AA-2M-tag against 3 *Escherichia coli* strains and clinically relevant pathogens, including *Pseudomonas aeruginosa*, *Salmonella* Typhimurium, *Klebsiella pneumoniae*, and *Acinetobacter baumannii*. The MICs of 92.5 to 370 μM were obtained against these strains and the $IC_{50}$ values were similar, ranging from 27.56 to 47.19 μM (Table 2 and S16 Fig). These results suggested that 19AA-2M-tag, a peptide derived from the colistin resistance gene, effectively combated antibiotic-resistant isolates. Interestingly, 19AA-2M-tag displayed synergetic activity with colistin (CT) against *E. coli* ATCC 25922 (FICI = 0.312) and a carbapenem-resistant *Enterobacteriaceae* (CRE) strain collected from the clinic (FICI = 0.156, S17 Fig). Moreover, the in vivo efficacy of the 19AA-2M-tag was demonstrated in mouse models of peritonitis against the *E. coli* ATCC 25922 and CRE clinical isolate. After 16 h of treatment, 19AA-2M-tag significantly reduced the bacterial burden in both liver and spleen compared to the untreated group (Fig 4E). Together, these results suggested that synthetic peptides derived from the M6 represent a potential strategy to combat drug-resistant bacteria both in vivo and in vitro.

## Discussion

In this study, we identified an MCR-1 mutant harboring 2 point mutations within the linker domain, causing significant disruptions in lipid A. Unlike MCR-1, the mutant protein confers phenotypic co-resistance to β-lactam antibiotics while inducing severe lipid A perturbations. This disorder eventually results in growth arrest, membrane permeabilization, and activation of LDTs pathway. Moreover, we have identified a lipid A binding pocket that is critical for

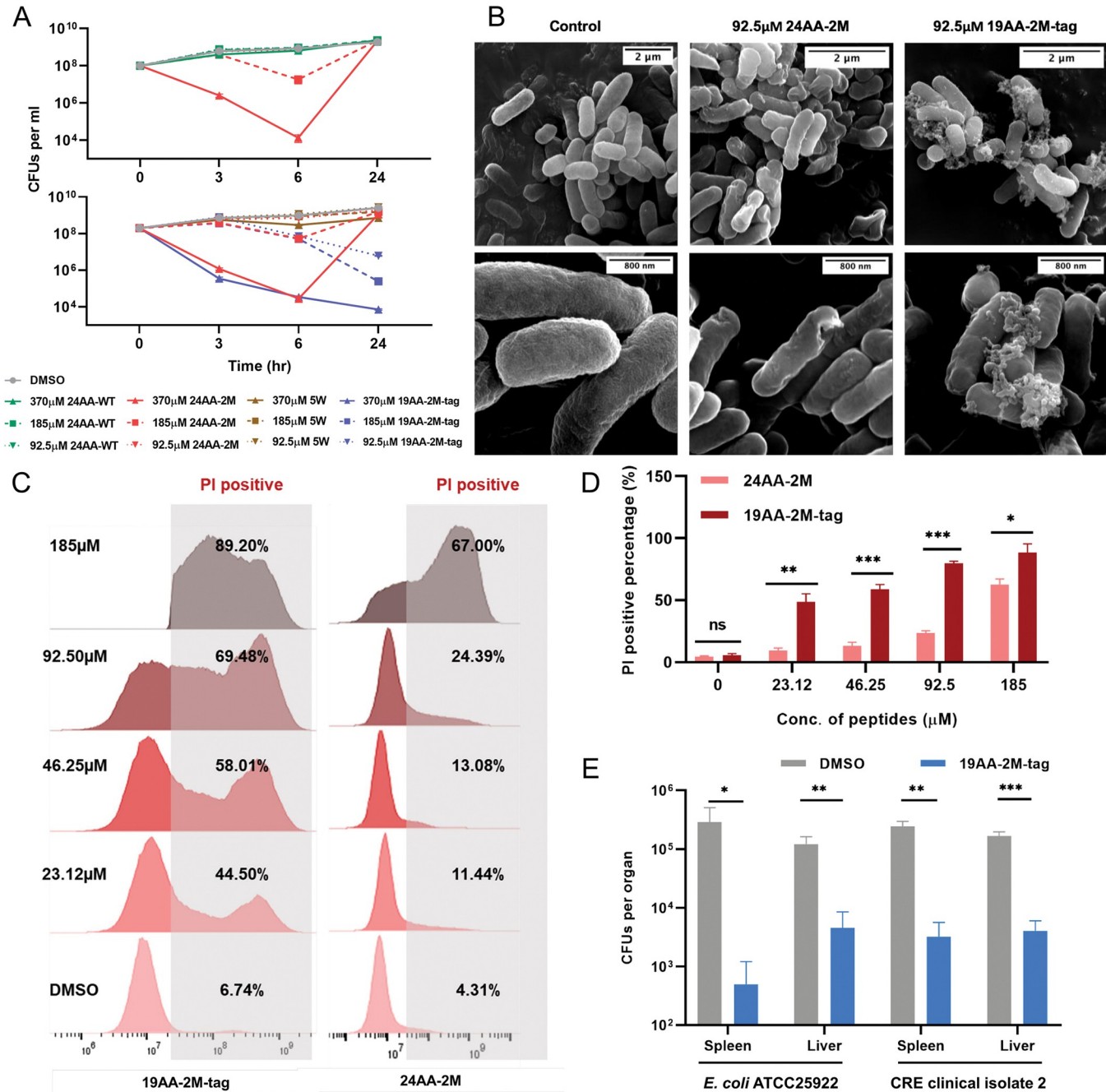

**Fig 4. Synthetic peptides derived from M6 displayed antimicrobial activity in vitro and in vivo. (A)** Time-dependent killing of *E. coli* ATCC 25922 by 24AA-WT, 24AA-2M, or 19AA-2M-tag. The exponential culture of *E. coli* ATCC 25922 was challenged with the indicated peptides. Cultures treated with DMSO or 5W were set as control. **(B)** SEM analysis of *E. coli* ATCC 25922 treated with 24AA-2M or 19AA-2M-tag in the concentration of 0.25× MICs. **(C, D)** Evaluation of the bacterial membrane permeability with the treatment of 24AA-2M or 19AA-2M-tag in vitro by PI staining assay. **(E)** Evaluation of the bactericidal ability of 19AA-2M-tag in vivo. C57BL/6 mice were first infected with $1 \times 10^7$ CFUs of *E. coli* ATCC 25922 or CRE isolated in the clinic through intraperitoneal injection for 1 h, followed by treatment with 200 μg of the indicated peptides. The infected individuals treated with DMSO were used as controls. Liver or spleen bacterial burdens were determined 16 h after treatment by spotting serial dilutions of tissue homogenates on LB plates. The average CFUs values of each organ with (blue) or without (gray) 19AA-2M-tag treatment were counted 16 h after incubation at 37°C. All the above-described experiments were performed thrice with similar results. Error bars indicate standard errors of the means (SEMs) for 3 biological replicates. A two-tailed unpaired *t* test was performed to determine the statistical significance of the data. *, $P < 0.1$; **, $P < 0.01$; ***, $P < 0.001$. The raw data underlying this figure can be found in S1 Data. CFU, colony-forming unit; CRE, carbapenem-resistant Enterobacteriaceae; LB, Luria–Bertani; PI, propidium iodide; SEM, scanning electron microscopy.

**Table 2. 19AA-2M-tag exhibits broad-spectrum antibacterial activity.**

| Strains | MICs (μM) |
| --- | --- |
| *Escherichia coli* BW25113 | 185 |
| *Escherichia coli* MG1655 | 370 |
| *Escherichia coli* ATCC 25922 | 370 |
| *Pseudomonas aeruginosa* ATCC 27853 | 185 |
| *Salmonella enterica serovar Typhimurium* SL1344 | 185 |
| *Klebsiella pneumoniae* ATCC 13883 | 185 |
| *Acinetobacter baumannii* ATCC 19606 | 185 |
| *A. baumannii* clinical isolate 1 | 185 |
| *A. baumannii* clinical isolate 2 | 185 |
| *E. coli* CRE clinical isolate 1 | 92.5 |
| *K. pneumoniae* CRKP clinical isolate 1 | 185 |
| *E. coli mcr-1*[+] clinical isolate 1 | 370 |
| *E. coli mcr-1*[+] clinical isolate 2 | 370 |

colistin resistance and bacterial membrane integrity. The mutated pocket in M6 exhibited enhanced affinity towards lipid A, potentially underpinning the β-lactams co-resistance phenotype. However, the most striking outcome of our research is that antimicrobial peptides derived from MCR-1 protein itself provide a new strategy to combat drug resistance.

Of the 10 *mcr* gene variants (*mcr-1* to *mcr-10*) identified to date [46,47], MCR-1 is considered the largest lineage with the highest prevalence, followed by MCR-3 [48,49]. Although a two-step enzymatic hydrolysis has been proposed, the exact mechanism governing the interaction between MCR-1 and lipid A remains unclear [50]. Recent research on an MCR-3 protein isolated from *Aeromonas hydrophila* suggests that a linker region of 59 residues (Linker 59) plays a crucial role in forming a phosphatidylethanolamine (PE) substrate-binding cavity and governs the interaction with lipid A, which is exposed to the bacterial periplasm [51]. In contrast, this study revealed a distinct lipid A binding pocket within the cytoplasmic membrane. Several lines of evidence support our hypothesis that the lipid A binding cavity of MCR-1 is situated within the linker domain. First, our previous work demonstrated that WT MCR-1 caused OM permeabilization, resulting in fitness costs for bacteria, particularly during the stationary growth phase. The increased OM permeability was directly related to the lipid A disturbance induced by MCR-1 [31]. Here, we have identified the MCR-1 variant M6, which causes unique phenotypic characteristics to bacteria, including growth retardation, up-regulation in LDTs pathway, and co-resistance to β-lactams. The phenotypic co-resistance induced by M6 was abolished by overexpressing LpxC, an enzyme responsible for the initial step of lipid A synthesis. These findings expose a link between membrane lipid A perturbation and phenotypic co-resistance induced by M6. Second, the location of the two-point mutations within the linker domain instead of the catalytic domain suggests that the effects induced by these mutations are not related with changes in the catalytic process. Third, combining MD simulation with AlphaFold structural predictions, we have confirmed the presence of a lipid A binding pocket within the linker domain of MCR-1. We also introduced mutations to interfere the interplay between such pocket and lipid A. Expression of these mutated proteins demonstrated a reduction in colistin resistance and restoration of bacterial membrane integrity. Moreover, a strong correlation between the membrane permeability and colistin resistance was observed among these mutants. Based on our observations, we propose an unusual "loading-transferring" mode for this enzyme (S18 Fig). This mechanism involves several key steps: during the first-half reaction, MCR-1 in apo state interacts with the PE donor in the outer

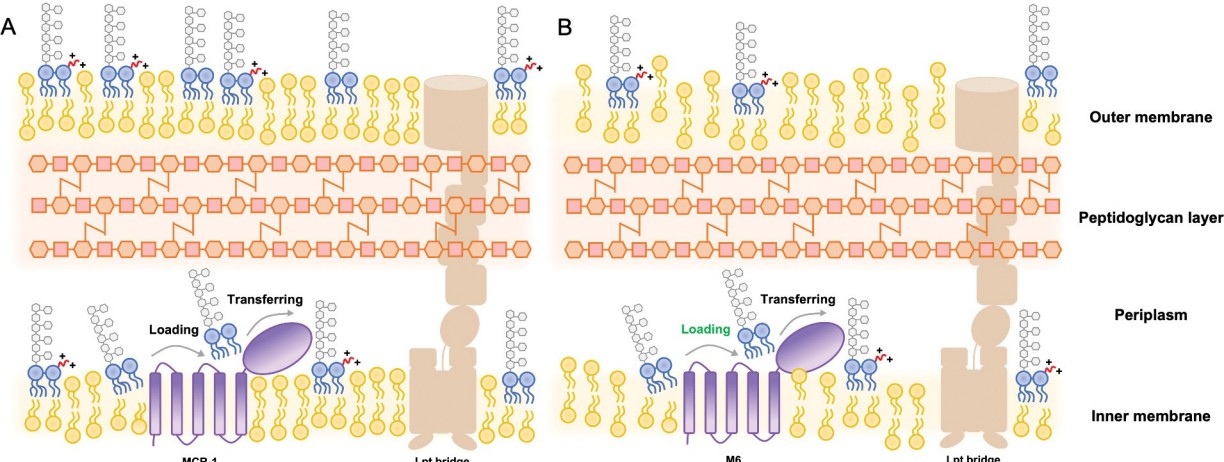

**Fig 5. Mode of membrane perturbation induced by MCR-1 or M6.** The lipid A PEA modification mediated by MCR-1 involves in following main steps: (i) lipid A interacts with the binding pocket within linker region, followed by (ii) transferring lipid A to the catalytic domain to accept PEA group. Finally, (iii) lipid A-PEA is released from MCR-1. (**A**) The expression of wild-type MCR-1 is able to modify LPS localized at the outer leaflet of IM. Subsequently, lipid A-PEA is transported to OM through Lpt bridge, a trans-envelope protein family for lipid A transportation. However, the expression of M6-induced perturbation upon membrane lipid A. (**B**) The enhanced lipid A substrate loading process of M6 (highlighted in green) appears to be responsible for the reduced lipid A on both IM and OM and, to some extent, reduces the bacterial sensitivity towards β-lactam antibiotics. IM, inner membrane; LPS, lipopolysaccharide; OM, outer membrane; PEA, phosphoethanolamine.

leaflet of the IM, accepting a PEA group in the catalytic domain. This step transforms MCR-1 into an active state and prepares it for interactions with LPS. Subsequently, LPS binds to the MCR-1 loading pocket within the linker domain, which is embedded in the lipid bilayer. Next, LPS is transferred to the catalytic domain, which is exposed at the interface between the IM and periplasm. During the second-half reaction, the PEA group from catalytic domain is added to either the 1′- or 4′-phosphate group of LPS, forming PEA-LPS, which is then released back into the IM. MCR-1 reverts to its apo state, ready for the next cycle of LPS modification.

We do not yet understand how MCR-1 impairs membrane integrity. It is well established that the cytoplasmic membrane protein PbgA plays a crucial role in sensing LPS levels and maintaining membrane LPS homeostasis [52]. Our research shown that MCR-1 expression leads to an increase in PbgA levels, indicating an aberrant distribution of LPS in *E. coli*. Consistently, a reduced level of LPS was observed in MCR-1 positive cells. We therefore concluded that the expression of MCR-1 increased bacterial permeability by reducing cellular LPS, which was enhanced by the expression of M6 and resulted in activation of LDTs pathway as well as remodeling of PG layer (Fig 5). However, the precise molecular mechanism by which MCR-1 impacts LPS transport remains unknown.

There are still several other considerations arising from our study. First, our findings indicate that severe membrane damage occurs by M6-induced activation of LDTs pathway, resulting in reduced susceptibility of *E. coli* to β-lactam antibiotics. Several studies have shown that PG remodeling mediated by the LDTs pathway is associated with increased OM permeability [36,39,53,54]. However, how M6 regulates this pathway merits further investigation. Second, the amino acid sequence of the linker domain varies among the 10 known MCR variants (S2 Fig). Different MCR variants exhibited varying levels of colistin resistance [55,56]. Our data revealed that the catalytic domain of the MCR proteins is relatively well conserved among the MCR families (S2 Fig), which strongly suggests that differences in the linker domain potentially govern the diverse phenotypic polymyxin resistance, an idea that will require direct experimental validation. Third, it appears that MCR-1 has pleiotropic effects and under

colistin selection, the acquisition of *mcr-1* guarantees a competitive advantage within bacterial communities. However, in the absence of colistin, the lipid A binding pocket of MCR-1, which disrupts membrane lipid A homeostasis, imposes a fitness cost on bacteria. Very recently, Pramod and colleagues [57] represented evidence that LpxC mutations compensated for the costs associated with MCR expression. Preexisting LpxC mutations in pathogenic *E. coli* can potentiate the evolution of antibiotic resistance by *mcr-1*-bearing plasmid acquisition [58]. Given the co-resistance phenotype conferred by the M6 mutant, this type of gain-of-function mutation could be concerning in the clinical setting.

Antibiotic treatment has given rise to the emergence of drug-resistant bacteria. Addressing the ongoing antibiotic crisis demands the development of newer strategies against drug-resistant infections. In our study, we identified a promising antimicrobial peptide derived from the drug resistance enzyme itself. Antimicrobial peptides (AMPs) are short peptides that tightly bind to the LPS on the membrane surface of bacteria. Such feature aligns with the high specificity of the lipid A binding motif found in transmembrane proteins, facilitating crucial protein-substrate interactions. A recent study shown that a synthetic peptide derived from the lipid A binding motif of PbgA displayed broad-spectrum antimicrobial activity [40]. Moreover, the introduction of an LPS binding motif into the C-terminus of temporin produces a broad spectrum of antibacterial activity [59]. Our findings extend the repertoire of AMP-based novel approaches to combat drug-resistant bacteria.

## Materials and methods

### Ethics statement

With the approval from Ethics Committee of Zhongshan School of Medicine on Laboratory Animal Care (reference number: SYSU-IACUC-2022-B0031), Sun Yat-sen University, all the animal experiments were conducted based on the standard of the National Institutes of Health Guide for the Care and Use of Laboratory Animals.

### Bacterial strains and growth conditions

The *E. coli* strains used in this research were ATCC 25922 and K-12 derivatives, including BW25113 and DH5-α. BW25113 cells were utilized to assay the influence of MCR-1 or M6 expression upon bacterial drug sensitivity, viability, and membrane permeability, while DH5-α cells acted as a cloning host for plasmid construction. The ATCC 25922 strain was routinely used to evaluate the in vitro bacteriostasis activity of synthetic peptides and murine in vivo infection. In addition, *Pseudomonas aeruginosa* ATCC 27853, *Salmonella enterica* serovar Typhimurium SL1344, *Klebsiella pneumoniae* ATCC 13883 as well as *Acinetobacter baumannii* ATCC 19606 were also used to test the susceptibility towards 19AA-2M-tag. All the above strains were cultivated in Luria–Bertani (LB) broth at 37˚C, antibiotics and promoter inducers were supplemented as follows when necessary: 100 μg/ml for ampicillin (AMP), 30 μg/ml for chloramphenicol (CHL), and 0.2% arabinose. All the strains used in this study were listed in S7 Table. For complete Materials and methods, see S1 Supplementary materials.

## Supporting information

**S1 Supplementary materials. Demonstration of Materials and methods in details.** (DOCX)

**S1 Fig. Identification of isolates with reduced sensitivity towards β-lactam antibiotics from the MCR-1 mutant library. (A)** The schematic illustration shows the process of screening MCR-1 mutants with reduced susceptibility towards β-lactams. The variant library

containing 171969 genotypes was cloned into the medium-copy plasmid pACYDuet-1 to generate the *E. coli* BW25113 strain pool. *E. coli* strains carrying WT MCR-1 or empty plasmid were set as control. Logarithmic-phase cultures of the 3 strains were first induced with 0.2% arabinose for 2 h, followed by plating on LB agar plates containing AMP, CAZ, IMP, SM, TET, NAL, or VAN. All antibiotics were in the concentrations of 0.8×, 1×, 1.5×, and 2× MICs. After incubation at 37˚C for 16 h, the number of surviving bacilli was counted to evaluate antibiotic susceptibility (**B**). It appears that certain MCR-1 variants exhibited reduced sensitivity to β-lactam antibiotics. To confirm the sensitivity of β-lactam antibiotics among the 3 strains, a similar process was performed, and well-induced cultures were plated on LB agar plates containing CTX, FEP, CRO, ETP, MEM, or FOX. CFUs were counted after incubation at 37˚C for 16 h (**C**). Fifty colonies of the *mcr-1* library in the background of *E. coli* BW25113 were selected from the plates containing CAZ or AMP. For each isolate, the pACYCDuet-1 plasmid carrying *mcr-1* variants was extracted, followed by transformation into *E. coli* BW25113 electroporation competent cells to eliminate influence caused by chromosomal mutation. Next, reconstructed strains were subjected to agar dilution MICs test to verify susceptibility to CAZ, AMP, or FOX. In addition, the *mcr-1* genotypes of the target isolates were verified through Sanger sequencing. Both panels (**B**) and (**C**) were visualized with Prism 9 software. All the above-described experiments were performed 3 times with similar results. Error bars indicate standard errors of the means (SEMs) for 3 biological replicates. The raw data underlying this figure can be found in S1 Data.
(PDF)

**S2 Fig. Conservation of the protein structure of the MCR family.** A surface representation shows the conservation analysis calculated across 106 MCR family proteins, in which red represents low homologous positions and green represents high conservation. The linker region and the mutated residues are shown in cartoon and sticky representations, respectively. The protein structure was analyzed and plotted by PyMOL 2.6 software.
(PDF)

**S3 Fig. Verification of the β-lactam antibiotic co-resistance rendered by M6. (A)** Zone of inhibition of *E. coli* BW25113 harboring empty plasmid, MCR-1 or M6 generated by disk diffusion method. (**B**) Different levels of antibiotic sensitivity in *E. coli* expressing either wild-type MCR-1, M6 or its single point mutants. The raw data underlying this figure can be found in S1 Data.
(PDF)

**S4 Fig. Co-resistance and membrane perturbation induced by native promoter-M6.** pACYCDuet-1 carrying MCR-1 or M6 under the regulation of the MCR-1 native promoter (NP MCR-1 or NP M6) was generated. Overnight cultures were subcultured into fresh LB broth at a ratio of 1:100. For the gene under the regulation of arabinose promoter, addition of 0.2% arabinose was required for protein expression induction. Logarithmic phase cultures were collected for following assays. (**A**) The sensitivity of indicated strains towards CT and β-lactam antibiotics (AMP, FOX, and CAZ) were evaluated by agar dilution MIC tests. Each triangle represents an independent experiment. The experiments were performed 3 times with similar results. NP, native promoter. (**B**) Efficiency of plating assays on LB agar plates containing 0.1% SDS and 1 mM EDTA. Ten-fold serial dilution of indicated cultures was inoculated onto the agar plates. (**C**) The OM integrity of indicated strains was determined by measuring NPN uptake. And the fluorescent signal for each sample was monitored with a microplate reader at an excitation wavelength of 350 nm and emission wavelength of 420 nm after staining. (**D, E**) The IM permeability of indicated strains was evaluated by PI staining assay.

Overnight cultures were subcultured into fresh LB broth at a ratio of 1:100. After cultivation for 8 h, cultures were collected, respectively, followed by staining with PI dye for 15 min. The PI-positive proportion was determined by flow cytometry and analyzed by FlowJo version 10 software. All the above-described experiments were performed thrice with similar results. Error bars indicate standard errors of the means (SEMs) for 3 biological replicates. A two-tailed unpaired *t* test was performed to determine the statistical significance of the data. ns, no significant difference; **, $P < 0.01$; ***, $P < 0.001$. The raw data underlying this figure can be found in S1 Data.
(PDF)

**S5 Fig. Expression of M6 inducing *E. coli* membrane shrinkage. (A)** Representative periplasmic GFP and cytoplasmic mCherry images of *E. coli* BW25113 cells expressing MCR-1 or M6 during stationary stage. Shrinkage of the cytoplasm was evident by the bright periplasmic GFP signal. **(B)** The percentage of cells exhibiting shrinkage at pole(s) was calculated. The graph was visualized with Prism 9 software. All the above-described experiments were performed 3 times with similar results. Error bars indicate standard errors of the means (SEMs) for 3 biological replicates. A two-tailed unpaired *t* test was performed to determine the statistical significance of the data. **, $P < 0.01$; ***, $P < 0.001$. The raw data underlying this figure can be found in S1 Data.
(PDF)

**S6 Fig. Membrane voltage measured by genetically encoded sensor Vibac2. (A)** The fluorescence ratio indicates the relative membrane voltage by using a genetically encoded voltage sensor Vibac2, which is a double-channel fusion protein that emits green and red fluorescence. The fluorescence intensity of GFP (Ig, Excitation = 488 nm, Emission = 512 nm) responds to membrane voltage, and the mCherry intensity (Ir, Excitation = 561 nm, Emission = 610) is used to normalize protein expression. Thus, the fluorescence ratio (Ig/Ir) indicates the relative membrane voltage in *E. coli* cells. To evaluate the impact of MCR-1 or M6 expression on the membrane voltage, a pBAD24 plasmid with the gene encoding Vibac2 was transformed into the indicated strains, and the fluorescence ratio (Ig/Ir) for each strain were measured and calculated during logarithmic phase. For each strain, 300 isolates were analyzed. And the representative GFP and mCherry images of BW25113 cells carrying empty plasmid, MCR-1 or M6 were represented as **(B)**. The raw data underlying this figure can be found in S1 Data.
(PDF)

**S7 Fig. Comprehensive proteome profiles of M6 using label-free quantitative proteomics. (A)** Venn diagram representing the differentially expressed proteins of *E. coli* BW25113 that harbored empty vector (control), MCR-1 and M6 during the exponential phase with the induction of 0.2% arabinose. **(B)** Top 20 pathways according to GO enrichment. The size and color of the points represent the number of target proteins and the q-values, respectively. The rich factor showed the enrichment degree in the GO pathway. **(C)** Visualization of the top KEGG enrichment pathway in M6-expressing cells compared with MCR-1-expressing cells. The up-regulated pathway is labeled in red, whereas the down-regulated pathway is labeled in blue. **(D)** Protein–protein interaction (PPI) analysis of the peptidoglycan layer remodeling pathway in M6 based on proteomics profiles, which were analyzed by STRING and visualized with Cytoscape software. Two biological replicates for each strain were used for the analysis.
(PDF)

**S8 Fig. β-Lactams and SDS susceptibility for LDTs-defective strains.** To test the role of LDTs in the M6-mediated phenotype, LdtD, PBP1B, or LpoB null strains carrying empty vector (control), WT MCR-1 or M6 were generated. **(A)** Antibiotic sensitivity of M6-expressing cells and MCR-1-expressing cells in the presence of copper. Overnight cultures of indicated

strains were subcultured into fresh LB broth with or without 3.75 mM $CuSO_4$ at a ratio of 1:100 and induced with 0.2% arabinose for 2 h. Next, the logarithmic-phase cultures were collected and adjusted to $OD_{600} = 0.6$, followed by spotting serial dilutions on LB agar plates containing CAZ or AMP, together with or without the addition of 3.75 mM $CuSO_4$. MICs were determined after incubation at 37˚C for 16 h. Each triangle represents an independent experiment. **(B)** Role of LDTs on β-lactam antibiotics susceptibility of M6. Overnight cultures of the indicated strains were subcultured into fresh LB broth at a ratio of 1:100 and induced with 0.2% arabinose for 2 h. The logarithmic-phase cultures were collected and adjusted to $OD_{600} = 0.6$, followed by spotting serial dilutions on LB agar plates with target antibiotics and incubation at 37˚C for 16 h. Each triangle represents an independent experiment. The experiments were performed 3 times with the same results. **(C)** Efficiency of plating assays on LB agar plates containing 0.1% SDS and 1 mM EDTA or 0.001% SDS and 1 mM EDTA. Ten-fold serial dilution of indicated cultures was inoculated onto the agar plates. **(D)** *mrcB* deletion failed to reverse the OM permeability defect caused by M6. NPN uptake is represented by the background subtracted fluorescence at an excitation wavelength of 350 nm and emission wavelength of 420 nm. **(E, F)** *mrcB* deletion abolished M6-mediated IM integrity. The IM permeability was evaluated by PI staining assay. Overnight cultures were subcultured into fresh LB broth at a ratio of 1:100 and induced with 0.2% arabinose to express WT MCR-1 or M6. After induction for 4 h and 8 h, stationary and late-stationary phase cultures were collected, respectively, followed by staining with PI dye for 15 min. The PI-positive proportion was determined by flow cytometry and analyzed by FlowJo version 10 software. Representative results of 3 independent experiments are shown in **(E)**. All the above-described experiments were performed 3 times with similar results. Error bars indicate standard errors of the means (SEMs) for 3 biological replicates. A two-tailed unpaired *t* test was performed to determine the statistical significance of the data. ns, no significant difference; *, $P < 0.1$; **, $P < 0.01$; ***, $P < 0.001$. The raw data underlying this figure can be found in S1 Data.
(PDF)

**S9 Fig. Spheroplasts LPS level perturbation of MCR-1 or M6-bearing *E. coli*. (A)** Morphology observation of spheroplasts for indicated strains. The shape of *E. coli* was changed from rob-shape into ball-shape, representing successful generation of spheroplasts. **(B, C)** The IM permeability of indicated spheroplasts was evaluated by PI staining assay. The well-prepared spheroplasts were incubated with PI dye for 15 min. The PI-positive proportion was determined by flow cytometry and analyzed by FlowJo version 10 software. All the above-described experiments were performed 3 times with similar results. Error bars indicate standard errors of the means (SEMs) for 3 biological replicates. A two-tailed unpaired *t* test was performed to determine the statistical significance of the data. ns, no significant difference; *, $P < 0.1$; ****, $P < 0.0001$. The raw data underlying this figure can be found in S1 Data.
(PDF)

**S10 Fig. Cellular LPS level in spheroplasts and LpxC overexpressing strains.** The cellular LPS level in spheroplasts **(A)** or LpxC overexpressing strains **(B)** harboring empty plasmid, MCR-1 or M6 were determined by western blot. Error bars indicate standard errors of the means (SEM) for triple biological replicates. A two-tailed unpaired *t* test was performed to determine the statistical significance of the data. *, $P < 0.1$; **, $P < 0.01$; ***, $P < 0.001$. The bar graph was visualized with Prism 9 software. The raw data underlying this figure can be found in S1 Data.
(PDF)

**S11 Fig. Necessity of P188-P195 segment for MCR-1 activity. (A)** Deletion of the region that includes P188-P195. *E. coli* BW25113 carrying ΔP188-P195 was generated, and the

susceptibility towards colistin (CT) and β-lactam antibiotics (AMP, FOX and CAZ) were evaluated by agar dilution MIC tests **(B)**. The experiments were performed 3 times with the same results. **(C)** Efficiency of plating assays on LB agar plates containing 1% SDS and 1 mM EDTA or 0.001% SDS and 1 mM EDTA. Ten-fold serial dilution of indicated cultures was inoculated onto the agar plates. The raw data underlying this figure can be found in S1 Data.
(PDF)

**S12 Fig. Mutations of essential regions for forming the lipid A binding cavity.** The close-up view shows the MCR-1 lipid A binding cavity with LPS and the 4 regions responsible for anchoring LPS. The linker domain and transmembrane domain are in magenta and orange, respectively. LPS binding with MCR-1 is represented as a stick mode and colored in green. The salt bridges for the interaction with LPS are shown in red. The change in structure with mutated residues in each region is shown in detail.
(PDF)

**S13 Fig. Impact of P188A+P195S mutations upon lipid A binding pocket. (A)** Represents the RMSD fluctuations for WT and M6 of MCR1-lipid A. The variations in MCR-1 MM-GBSA binding energy with WT and M6 during the MD trajectories are shown in **(B). (C, D)** The statistics for the observed molecular interactions between lipid A and MCR-1 over the simulated trajectories of WT and the M6 mutant (the last 200-ns trajectories used for binding free energy calculations). **(E)** Quantification of lipid A with or without the modification of phosphoethanolamine (PEA). The bar graph represented the abundance of modified lipid A and unmodified lipid A extracted from the whole cells and spheroplasts of *E. coli* BW25113 expressing MCR-1 or M6, which was determined by MALDI-TOF. The experiments were performed 3 times with similar results. The raw data underlying this figure can be found in S1 Data.
(PDF)

**S14 Fig. Localization of FITC-labeled peptides in *E. coli* ATCC 25922.** Logarithmic phase culture of *E. coli* ATCC 25922 was collected and stained by FM4-64 (red) and DAPI (cyan) to indicate the localization of bacterial membrane as well as cytoplasmic chromosome. The well-stained culture was subsequently treated with FITC-labeled 24AA-2M and 19AA-2M-tag in the concentrations of 0.05× and 0.1× MICs (18.5 μM and 37.0 μM), respectively. After washing with 1× PBS, the localization of the FITC signal were determined by observation with fluorescent microscope.
(PDF)

**S15 Fig. Permeabilization of synthetic peptides on mouse blood cells.** To identify the cytotoxicity of 24AA-2M and 19AA-2M-tag, the cell permeabilizing effects of the indicated peptides on mouse blood cells were determined by an LDH-based TOX-7 kit (Sigma). Fresh healthy mouse blood was treated with the above peptides in the concentrations of 46.25, 92.5, 185, or 370 μM. LDH activity was evaluated to determine LDH released from mousse cells. Error bars indicate standard errors of the means (SEMs) for 3 biological replicates. A two-tailed unpaired *t* test was performed to determine the statistical significance of the data. ns, no significant difference; *, $P < 0.1$; **, $P < 0.01$. The bar graph was visualized with Prism 9 software. N.D., not detected. The raw data underlying this figure can be found in S1 Data.
(PDF)

**S16 Fig. Inhibitory rate of 19AA-2M-tag among gram-negative strains.** By treating *E. coli* ATCC 25922, *P. aeruginosa* ATCC 27853, *S. Typhimurium* SL1344, *K. pneumonic* ATCC 13883, and *A. baumanii* ATCC 19606 with 19AA-2M-tag in a series of concentrations, dose-

response curves were generated. The in vitro inhibitory rate and $IC_{50}$ value of various strains treated with 19AA-2M-tag are presented. The raw data underlying this figure can be found in S1 Data.
(PDF)

**S17 Fig. Synergistic inhibition of *E. coli* growth by 19AA-2M-tag and colistin. (A)** *E. coli* ATCC 25922 and a carbapenem-resistant *E. coli* isolated in the clinic were used to test the efficacy of a drug combination consisting of 19AA-2M-tag and colistin through a checkerboard assay. A total of $1 \times 10^3$ CFUs of the indicated strains were inoculated at initiation (T0), followed by treatment with drug combinations in a series of concentrations at 37˚C for 16 h (Tn). The $OD_{600}$ was measured before and after the treatment. The graphs show the value of $OD_{600\,Tn}$ minus the $OD_{600\,T0}$ of each well for both strains. A color gradient heat map, with hot (red: $OD_{600}$ = 0) to cool (blue: $OD_{600}$ = 0.5) colors, indicates low to high values. The synergetic effect was further determined by calculating the FICI. **(B)** Shows the inhibition curves of colistin with (red) or without (blue) the addition of 19AA-2M-tag. The dose-response curves of *E. coli* ATCC 25922 were generated by measuring the inhibitory rate after treatment with colistin in a series of concentrations, with or without co-treatment with 92.5 μM 19AA-2M-tag. The raw data underlying this figure can be found in S1 Data.
(PDF)

**S18 Fig. The "loading-transferring" mode of MCR-1.** LPS modification mediated by MCR-1 might be guaranteed by a two-step process, which requires the LPS substrate to be loaded into binding pocket and transferred to the catalytic domain. During the first-half reaction, the Apo state MCR-1 interacts with the PE donor at the outer leaflet of the IM and accepts a PEA group in the catalytic domain, transforming into the active state to interact with LPS. Next, LPS binds with the MCR-1 loading cavity at the linker domain, which is inserted into the lipid bilayer, followed by transfer to the catalytic domain exposed at the interface between the IM and periplasm. During the second-half reaction, the PEA group at the catalytic domain was added to the 1′- or 4′-phosphate group of LPS to form PEA-LPS, which was then released back into the IM, and MCR-1 switched to the Apo state again.
(PDF)

**S19 Fig. Plasmid maps of pACYC-Para-MCR-1 and pACYC-NP-MCR-1.** The plasmid for expressing MCR-1 or M6 under the regulation of arabinose promoter was shown as **(A)**, and the one for expressing target proteins under the regulation of MCR-1 native promoter was shown as **(B)**. The elements in each plasmid were as followed: CmR = chloramphenicol acetyltransferase; p15A *ori* = the medium-copy-number p15A origin of replication; T7 terminator = transcription terminator for bacteriophage T7 RNA polymerase; *araC* = gene encoding L-arabinose regulatory protein; *araBAD* promoter = promoter of the L-arabinose operon; MCR-1 native promoter = the native promoter of *mcr-1* gene amplified from clinically collected *mcr-1*[+] strain; WT MCR-1/M6 = the gene encoding WT MCR-1 or M6.
(PDF)

**S20 Fig. Analysis of the molecular dynamic simulation and binding energy.** Molecular dynamic stimulation was utilized to discover the potential LPS binding cavity on the surface of MCR-1, and the workflow was designed as shown in **(A)**. Structural comparison within lipid A (PDB code: 5IJD) and its analogues: Eritoran (PDB code: 2Z65) and Palmitoyllpid A (PDB code: 7BGL) are shown in **(B)**. **(C)** The potential lipid A binding site on the surface of MCR-1 was discovered by SiteMap of Maestro. **(D)** Shows the RMSD fluctuations for multiple trajectories of MCR1-lipid A.
(PDF)

**S1 Table. Concentrations of antibiotics used for screening the MCR-1 library.**
(PDF)

**S2 Table. Concentrations of β-lactam antibiotics used for screening the MCR-1 library.**
(PDF)

**S3 Table. Verification of MCR-1 isolates through Sanger sequencing.**
(PDF)

**S4 Table. Differentially expressed proteins between *E. coli* BW25113 carrying M6 and WT MCR-1.**
(PDF)

**S5 Table. The significantly enriched GO terms of differentially expressed proteins between *E. coli* BW25113 carrying WT MCR-1 and M6.**
(PDF)

**S6 Table. The KEGG pathways map of differentially expressed proteins between *E. coli* BW25113 carrying WT MCR-1 and M6.**
(PDF)

**S7 Table. Bacterial strains used in this study.**
(PDF)

**S8 Table. Plasmids used in this study.**
(PDF)

**S9 Table. Primers used in this study.**
(PDF)

**S10 Table. Synthetic peptides used in this study.**
(PDF)

**S1 Data. Raw data underlines the results in this research.**
(XLSX)

**S1 Raw images. Raw images of western blot analysis performed in the research.**
(DOCX)

**S1 Supplementary PDB file. The predicted complex structure of MCR-1 and Lipid A.** The PDB file reveals the structure of MCR-1-lipid A complex. The catalytic domain, linker domain, and transmembrane domain are in cyan, magenta, and orange, respectively, with LPS is shown as green sticks.
(PDB)

## Acknowledgments

The plasmid encoding bacterial membrane voltage sensor Vibac2 was a generous gift from Prof. Bai Fan's lab.

## Author Contributions

**Conceptualization:** Lujie Liang, Lin Wang, Dianrong Zhou, Nicole Stoesser, Yohei Doi, Fang Bai, Guo-Bao Tian.

**Data curation:** Lujie Liang, Lin Wang.

**Formal analysis:** Lujie Liang, Lin Wang, Jiachen Li.

**Funding acquisition:** Siyuan Feng, Guo-Bao Tian.

**Investigation:** Lujie Liang, Dianrong Zhou, Yaxin Li, Yong Chen, Wanfei Liang, Wenjing Wei, Chenchen Zhang, Lingxuan Lyu.

**Methodology:** Lujie Liang, Lin Wang, Dianrong Zhou, Yaxin Li, Yong Chen, Wanfei Liang, Wenjing Wei, Chenchen Zhang, Lingxuan Lyu.

**Project administration:** Lujie Liang, Lin Wang.

**Resources:** Lan-Lan Zhong, Hui Zhao.

**Software:** Lin Wang.

**Visualization:** Lujie Liang, Lin Wang.

**Writing – original draft:** Lujie Liang.

**Writing – review & editing:** Lujie Liang.

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
