## [Editor Report · Decision Letter 0]

13 Apr 2023

Dear Dr. Tian, 

Thank you for submitting your manuscript entitled "Identification of a novel MCR-1 variant displaying low level of co-resistance to β-lactam antibiotics that uncovers a potential novel antimicrobial peptide" for consideration as a Research Article by PLOS Biology.

Your manuscript has now been evaluated by the PLOS Biology editorial staff, as well as by an academic editor with relevant expertise, and I am writing to let you know that we would like to send your submission out for external peer review.

Once your full submission is complete, your paper will undergo a series of checks in preparation for peer review. After your manuscript has passed the checks it will be sent out for review. To provide the metadata for your submission, please Login to Editorial Manager (https://www.editorialmanager.com/pbiology) within two working days, i.e. by Apr 15 2023 11:59PM.

Kind regards,

Paula

---

Senior Editor

PLOS Biology

---

## [Decision Letter · Decision Letter 1]

22 Jun 2023

Dear Dr. Tian,

Thank you for your patience while your manuscript "Identification of a novel MCR-1 variant displaying low level of co-resistance to β-lactam antibiotics that uncovers a potential novel antimicrobial peptide" was peer-reviewed at PLOS Biology. It has now been evaluated by the PLOS Biology editors, an Academic Editor with relevant expertise, and by several independent reviewers. 

In light of the reviews, which you will find at the end of this email, we would like to invite you to revise the work to thoroughly address the reviewers' reports.

As you will see below, the reviewers find your work interesting, but raise some concerns that will need to be solved addressed publication. In particular, we think it is very important that you address reviewer #1's concern regarding that the claims presented in the model in Figure 5 are not sufficiently supported by the data and the reviewer asks for further experimental validation that modified lipid A accumulates in the inner membrane when the M6 MCR-1 variant is expressed. Reviewer #2 notes the reliance on the Mcr-1 overexpression system and asks that the authors provide evidence that the same phenotypes are observed in conditions of physiological Mcr-1 expression.

Given the extent of revision needed, we cannot make a decision about publication until we have seen the revised manuscript and your response to the reviewers' comments. Your revised manuscript is likely to be sent for further evaluation by all or a subset of the reviewers.

**IMPORTANT - SUBMITTING YOUR REVISION**

*Re-submission Checklist*

*Published Peer Review*

*PLOS Data Policy*

*Blot and Gel Data Policy*

Sincerely,

Paula

---

Senior Editor

PLOS Biology

REVIEWS:

Reviewer #1: Bacteria resistance to antibiotics.

Reviewer #2: Antimicrobial resistance and antimicrobial peptides.

Reviewer #1: Liang et al describe a point variant of the MCR-1 lipid A phosphoethanolamine transferase whose expression also reduces susceptibility to beta-lactam antibiotics. A screen of a previously described MCR mutant library identified a dual variant, M6, with reduced beta-lactam susceptibility. NPN and propidium iodide staining experiments identified that the outer membrane permeability of M6-expressing cells was similar to that of those expressing wild-type MCR-1, but that M6 reduced effects upon the inner membrane compared to those arising from wild-type MCR-1 expression. Proteomic comparison of M6- and wild-type MCR-1-expressing E. coli identified that the former induced upregulation of genes involved in peptidoglycan biosynthesis and remodeling. Increased beta-lactam MICs in the M6 strain were shown to be LdtD and PBP1b-dependent, consistent with a mechanism involving 3-3 rather than 4-3 cross-linking. The region of MCR-1 disrupted in the M6 mutant was identified as a putative lipidA binding site, as evidenced by docking experiments and molecular dynamics simulations; and by the effects of deletion of this region or mutation at positions predicted to be involved in lipidA binding. Peptides based upon the putative lipidA binding site of the M6 variant, but not wild-type MCR-1, showed antibacterial activity in in vitro assays against several Gram-negative species, and against E. coli in a mouse infection model. It is concluded differential effects upon lipidA binding in the M6 mutant, compared to wild-type MCR-1, ultimately affect LPS levels/distribution and peptidoglycan composition. Furthermore, the authors suggest that the lipidA binding properties of M6 are exploitable in new generations of antimicrobial peptides.

The manuscript contains very substantial new data providing new information on the relationship between MCR expression and the organization of the bacterial envelope, and is clearly of considerable interest in the area of Gram-negative bacterial physiology as well as antimicrobial (colistin) resistance. I would however suggest some revision and clarification before recommending acceptance.

Most importantly, the connection between the data presented and the model proposed in Figure 5 is insufficiently clear. While I appreciate the uncertainties surrounding the connection between lipidA levels/modifications and the extent and type of peptidoglycan cross-linking, the reasoning underlying the model in Figure 5 is not explained either in lines 420 - 422 or in the legend. The Figure suggests that accumulation of modified lipidA in the inner membrane is occurring when the M6 variant is expressed, compared to wt MCR-1. Did the authors attempt to verify this experimentally, e.g. by mass spectrometry of spheroplasts from M6-expressing cells? It is then unclear how this relates to the reduction in PI accumulation in M6-expressing cells, compared to those expressing wild-type MCR-1, shown in Figure 1G.

Other comments:

Line 121: this section is unclear. Do the authors mean a defect in OM permeability (as written)? Ref. 30 suggests that what is meant is an increase.

Figure legends throughout lack detail. It should be possible to interpret Figures based upon the legend and without extensive reference to the main text. Figures 1B, 1E, 2F, 2G and 3D, E, F, G and H are just some of multiple examples where the legend lacks sufficient detail to make clear exactly what is being shown.

Line 200: clarify that while M6 expression clearly disrupts the OM this is also observed for wt MCR-1

Lines 256 -7 and 275 - 6: While I agree that the observed increase in LdtD expression, and the effects of LdtD inhibition, are consistent with beta-lactam resistance in the M6-expressing strains resulting from LdtD-mediated PG remodeling, in the absence of a direct determination of PG composition this conclusion cannot be regarded as conclusive. I would suggest that the relevant sections be reworded to reflect this.

Lines 262 - 264: were changes in PbgA expression evident from the proteomics experiments?

Figure 3C axis labels: surely NPN uptake measures OM, and PI IM, integrity, and not the reverse as shown?

Lines 317 - 319 and Figure 3: Which peptides are being used in these experiments? These should be detailed in the main text, as well as Methods.

lines 320 - 22: This is unclear. Is membrane potential what is being measured here?

Line 372: Can the authors argue that M6 induces OM permeabilization compared to wt MCR-1? This does not appear to be supported by the NPN uptake measurements in Figure 1E.

Reviewer #2: This manuscript by Tian and collaborators identifies a potential mechanism explaining membrane changes and fitness cost caused by the expression of the plasmid encoded Mcr1 ethanolamine transferase enzyme, which is responsible for inducing colistin resistance in Gram-negative bacteria. The authors describe a Mcr1 variant (M6) that causes outer membrane perturbation, increased resistance to beta-lactam antibiotics (co-resistance) and modifications in the cell wall peptidoglycan. The authors propose, based on modelling that the region modified in the M6 variant forms a cavity that accommodates lipid A and perturbs the affinity of the interaction, which in turn leads to alteration of lipid A levels and downstream effects related to cell permeability and PG defects. The authors also designed a peptide based on the linker sequence that has antimicrobial activities possibly by affecting lipid A. Overall, this manuscript has interesting results that advance the field.

I have several major concerns that require attention:

1. The work is based on the overexpression of the Mcr1 protein and its derivatives. pACYCDuet is a plasmid of medium copy number (~20-50 copies per chromosome) and the cloned genes are expressed by a T7 promoter, which is highly efficient. These levels of expression are very high relative to the natural expression of Mrc-1. Because Mrc1 is a membrane protein and membrane proteins are well known to induce pleiotropic effects when expressed, it would be important that the observed phenotypes are replicated under conditions of Mcr1 natural expression. 

2. Can the authors attribute the antimicrobial effects of the peptides to the identified mechanism of the linker region binding to lipid A. The peptides are provided to cells at micro-molar concentrations, which are likely not achievable clinically. Also, how do they enter bacterial cells? Is the entry of the peptides related to the membrane integrity defect induced by Mcr1 (or M6)? or they are inherently permeable to the OM? (Unlikely given their mass)

3. I find the manuscript confusing in several sections (see below). 

Other concerns:

L88-89, Is the topology of the ethanolamine transfer known? According to the sentence, the modification occurs after lipid A flipping across the inner membrane at the periplasmic side of the membrane. If this is true, the PEt should also be flipped across the membrane. Can the authors clarify?

L110, the lipid A core is transported to the outer leaflet of the OM and the consequence of this is the formation of an asymmetrical OM lipid bilayer. But the transport is not set to "establish OM asymmetry". Asymmetry also depends on the movement of phospholipid from the outer leaflet of the OM to the cell membrane by the Mla system. Please clarify sentence.

L114-117, Mcr1 is a membrane protein; many membrane proteins alter bacterial fitness when overexpressed. Is the fitness defect detectable when Mcr1 is expressed under normal conditions?

L127-128, according to the results in Fig 1, especially Fig1B, the M6 variant confers more resistance to beta-lactams and other antibiotics.

L131-132, revise the sentence; it is confusing.

L142-144, this needs to be explained better: the authors report a library of 171,169 mutant genotypes? I suppose they mean amino acid replacement mutants. 

L146-160, the screening needs to be better explained since the text is confusing. The authors appeared to have used a library of mutant plasmids and select for variants that confer resistance to beta-lactams. Further sequencing identified M6; is this correct?

L161-168, another confusing section: the authors show that M6 mutant is more resistant to the beta-lactams, also in Table S3, but in the text they say in L165 "lower sensitivity".

L171-186, the permeability assays should be done with the genes expressed at its normal dosage. pACYC184 is a plasmid of moderate copy number and the authors are consequently overexpressing a membrane protein.

L181-182, NPN uptake. The uptake is increased in the Mcr1 and M6 to high levels compared to the plasmid control, but the PI uptake indicate differences between Mcr1 and M6. These results may reflect a limitation of the NPN assay to identify differences at higher levels of fluorescence absorption. Was the assay linear?

L188-189, Why the authors conclude M6 confers a low level of co-resistance? They have shown that resistance to beta-lactams is increased?

L265-267, is the OM permeability increased or decreased?

L346, the authors tested the susceptibility of the strains to the peptide, not the other way around.

Suggested editorial changes.

Below is a list of the most obvious edits required. However, overall, the manuscript is verbose and requires careful editing to improve clarity and readability. 

L80, L120, L217 delete "Notably," here and throughout the manuscript.

L87, replace "bound" by "embedded."

L88, delete 

L113, replace "impact", by "affects"

L118, delete "Subsequently,"

L131, delete "Strikingly,"

L144-145, delete "To decrease..." until the comma and start with "BW25113..."

L173-174, replace "To do it", by "To confirm this"

L196, replace "target" by "selected"

L232, delete "kind of"

L240, replace "the survive" by "survival"

---

## [Decision Letter · Decision Letter 2]

1 Nov 2023

Dear Dr Tian,

Thank you for your patience while we considered your revised manuscript "Identification of a novel MCR-1 variant displaying low level of co-resistance to β-lactam antibiotics that uncovers a potential novel antimicrobial peptide" for publication as a Research Article at PLOS Biology. This revised version of your manuscript has been evaluated by the PLOS Biology editors, the Academic Editor and one of the original reviewers.

Based on the review and our Academic Editor's assessment of your revision, we are likely to accept this manuscript for publication, provided you satisfactorily address the remaining points raised by the reviewer and the following data and other policy-related requests.

IMPORTANT - please attend to the following:

a) For our broad readership, please change your Title to "A new variant of the colistin resistance gene MCR-1 with low level of co-resistance to β-lactam antibiotics reveals a potential novel antimicrobial peptide"

b) Please provide a blurb, according to the instructions in the submission form.

c) Please address the remaining comments from reviewer #2. Note that this reviewer has attached an annotated version of your manuscript, which you may find helpful.

d) Please rename your data file "Raw data 20230918.xlsx" to "S1_data.xlsx" and cite the location of the data clearly in all relevant main and supplementary Figure legends, e.g. “The data underlying this Figure can be found in S1 Data.”

e) Please make any custom code available, either as a supplementary file or as part of a deposition.

We expect to receive your revised manuscript within two weeks. 

*Published Peer Review History*

*Press*

Sincerely,

Roli Roberts

Roland G Roberts, PhD

Senior Editor

PLOS Biology

rroberts@plos.org

on behalf of

Editor,

pjaureguionieva@plos.org,

PLOS Biology

CODE POLICY

Per journal policy, as the code that you have generated is important to support the conclusions of your manuscript, we require that you make it available without restrictions upon publication. Please ensure that the code is sufficiently well documented and reusable, and that your Data Statement in the Editorial Manager submission system accurately describes where your code can be found.

SPECIES INDICATED IN THE ABSTRACT? 

- Please note that per journal policy, the model system/species studied should be clearly stated in the abstract of your manuscript. 

We require the original, uncropped and minimally adjusted images supporting all blot and gel results reported in an article's figures or Supporting Information files. We will require these files before a manuscript can be accepted so please prepare and upload them now. Please carefully read our guidelines for how to prepare and upload this data: https://journals.plos.org/plosbiology/s/figures#loc-blot-and-gel-reporting-requirements

DATA NOT SHOWN?

REVIEWER'S COMMENTS:

Reviewer #2:

IMPORTANT: ALSO SEE THE ATTACHED EDITED VERSION OF THE MANUSCRIPT

The revised manuscript of Liang et al. is considerably improved. The authors have responded extensively to the referees' comments and provided new experimental information. However, the manuscript continues to be poorly written. It is not only about English usage but mainly about style. On this respect, the style is verbose, and many sentences begin with "Notably", "remarkably", "unexpectedly" which make reading fastidious and are not necessary. 

Further there are several sentences that need major editing for clarity. For example, "two-points mutation" means nothing. The authors are referring to "two point mutations". 

In other cases, additional explanations are given when the references are provided, which could be safely deleted. 

There also confusion between genes mcr-1 (in italics) and the protein, MCR-1 (which technically should be referred to as Mcr-1 in keeping to the conventional naming of bacterial proteins). Also, in several occasions, the authors attribute to M6 an activity that is produced by the bacteria expressing this protein. 

Finally, some adjectives, like "severe" are subjective. 

There is a fair bit to rewrite but considering that English is not the mother tongue of the authors, I have taken the liberty to provide an edited copy for the authors' consideration (uploaded with this review), hoping it can help them to rework the manuscript and bring it to an adequate standard. I have not edited the figure legends, but they also need some work.

---

## [Editor Report · Decision Letter 3]

14 Nov 2023

Dear Dr. Tian,

Thank you for the submission of your revised Research Article "A new variant of the colistin resistance gene MCR-1 with co-resistance to β-lactam antibiotics reveals a potential novel antimicrobial peptide" for publication in PLOS Biology. On behalf of my colleagues and the Academic Editor, Csaba Pal, I am pleased to say that we can in principle accept your manuscript for publication, provided you address any remaining formatting and reporting issues. These will be detailed in an email you should receive within 2-3 business days from our colleagues in the journal operations team; no action is required from you until then. Please note that we will not be able to formally accept your manuscript and schedule it for publication until you have completed any requested changes.

PRESS

Sincerely, 

Paula

---

Senior Editor

PLOS Biology
